# Headspace Volatiles and Endogenous Extracts of *Prunus mume* Cultivars with Different Aroma Types

**DOI:** 10.3390/molecules26237256

**Published:** 2021-11-30

**Authors:** Xueqin Wang, Yanyan Wu, Huanhuan Zhu, Hongyan Zhang, Juan Xu, Qiang Fu, Manzhu Bao, Jie Zhang

**Affiliations:** Key Laboratory of Horticultural Plant Biology (Ministry of Education), College of Horticulture and Forestry Sciences, Huazhong Agricultural University, Wuhan 430070, China; wangxueqin@webmail.hzau.edu.cn (X.W.); wuyanyan@webmail.hzau.edu.cn (Y.W.); bucket@webmail.hzau.edu.cn (H.Z.); zhanghy@mail.hzau.edu.cn (H.Z.); xujuan@mail.hzau.edu.cn (J.X.); fuqiang@mail.hzau.edu.cn (Q.F.); mzbao@mail.hzau.edu.cn (M.B.)

**Keywords:** *Prunus mume*, headspace volatiles, endogenous extracts, HS-SPME, OSE, GC-MS, aroma types

## Abstract

*Prunus mume* is a traditional ornamental plant, which owed a unique floral scent. However, the diversity of the floral scent in *P. mume* cultivars with different aroma types was not identified. In this study, the floral scent of eight *P. mume* cultivars was studied using headspace solid-phase microextraction (HS-SPME) and organic solvent extraction (OSE), combined with gas chromatography-mass spectrometry (GC-MS). In total, 66 headspace volatiles and 74 endogenous extracts were putatively identified, of which phenylpropanoids/benzenoids were the main volatile organic compounds categories. As a result of GC-MS analysis, benzyl acetate (1.55–61.26%), eugenol (0.87–6.03%), benzaldehyde (5.34–46.46%), benzyl alcohol (5.13–57.13%), chavicol (0–5.46%), and cinnamyl alcohol (0–6.49%) were considered to be the main components in most varieties. However, the volatilization rate of these main components was different. Based on the variable importance in projection (VIP) values in the orthogonal partial least-squares discriminate analysis (OPLS-DA), differential components of four aroma types were identified as biomarkers, and 10 volatile and 12 endogenous biomarkers were screened out, respectively. The odor activity value (OAV) revealed that several biomarkers, including (*Z*)-2-hexen-1-ol, pentyl acetate, (*E*)-cinnamaldehyde, methyl salicylate, cinnamyl alcohol, and benzoyl cyanide, contributed greatly to the strong-scented, fresh-scented, sweet-scented, and light-scented types of *P. mume* cultivars. This study provided a theoretical basis for the floral scent evaluation and breeding of *P. mume* cultivars.

## 1. Introduction

The flower scent is an important trait of ornamental plants that affects the commercial value of flowers. Meanwhile, it attracts specific pollinators and seed communicators. In some cases, it also plays a role in plant defense [1]. Floral metabolites are widely used in perfumes, cosmetics, foods, and medical applications [2]. It was known that the floral scent is a complex of volatile organic compounds (VOCs) [3]. VOCs are composed of a series of lipophilic liquids with low molecular weight, low polarity, and high vapor pressure [4]. The VOCs in plants can be categorized into three major categories: phenylpropanoids/benzenoids, terpenoids, and fatty acid derivatives [4,5,6]. Up to now, more than 1700 volatiles have been identified from over 90 plant families [7]. In addition, the biosynthetic and release pathways of VOCs include a variety of intermediate products. A previous study revealed that endogenous substances were the basis of material volatilization [8]. Before volatilization, the VOCs are produced in liquid form and stored in the tissues of flowers. Due to the change of vapor pressure, a small number of compounds emitted from flowers and collected by the headspace method are called headspace volatiles. Most of the compounds that remained in flower tissues and were extracted by solvent are called endogenous extracts [3,9]. It was shown that the diversity of floral scents is caused by the changes of endogenous extracts content and the evaporation of headspace volatiles [3]. Therefore, it is necessary to analyze the headspace volatiles and endogenous extracts independently so as to obtain the relative contribution to the final aroma components [3,10,11].

There are three methods to extract the volatile compounds, such as extraction, distillation, and headspace methods [12]. At present, most of the volatile compounds in ornamental flowers are determined by headspace solid-phase microextraction (HS-SPME), including *Malus ioensis* [13], *Chrysanthemum* [14], and *Iris* [2]. HS-SPME can accurately analyze the volatile samples, enrich and separate the aroma components with a low threshold, which is more specific, simple and reliable to living plants. Solvent extraction is used to extract effective components from the raw materials according to their solubility in different solvents. This is suitable to study the endogenous components of the petals, such as *Petunia axillaris* [3], *Jasminum* [9], and *Dianthus* [10]. At present, GC-MS is often used to obtain chromatograms and mass spectra of VOCs [15,16].

*Prunus mume* (mei) is a traditional flower in China that has a long cultivation history and belongs to the *P*. genus of the Rosaceae family [8,11,17]. *P. mume* is native to the Yangtze River Basin in southwestern China and is also widely planted in northern China. It usually blooms from January to February, and it is a famous traditional ornamental plant in China [17,18]. Compared to other species of *Prunus*, mei has a unique floral scent [19]. Previous studies have already focused on the main volatile components, including benzyl acetate and eugenol, and their synthesis-related genes in mei. For instance, Zhao et al. [20,21,22] mainly concerned the volatile components of different *P. mume* cultivars and their temporal and spatial patterns. Benzyl acetate and eugenol, the metabolites of the phenylpropanoids/benzenoids metabolism pathway, were found to be the main compounds of the emitted and endogenous floral scent compounds of *P. mume* and hybrids [11]. Additionally, promoting and interfering with the expression of *PmBEAT34*, *36* and *37* could regulate the production of benzyl acetate in the petal protoplasts of *P. mume* [17]. Additionally, Zhang et al. found that *PmCFAT1* contributed to eugenol biosynthesis in *P. mume* [23]. However, little is known about the diversity of the floral scent in the *P. mume* cultivars with different aroma types and the difference in composition between headspace volatiles and endogenous extracts.

In this study, the headspace volatiles and endogenous extracts of eight *P. mume* cultivars with four aroma types were analyzed by headspace solid-phase micro-extraction (HS-SPME) and organic solvent extraction (OSE) combined with gas-chromatography mass-spectrometry (GC-MS), respectively. Then, the variable importance in projection (VIP) values were used to screen the differential compounds among the four aroma types as biomarkers. In addition, further analysis of odor activity value (OAV) screened out several biomarkers that made great contributions to the four aroma types. This study lays a foundation for the classification and breeding of *P. mume* cultivars in the future.

## 2. Results

### 2.1. Identification and Relative Content Analysis of the Composition of the P. mume Cultivars with Four Aroma Types

The components of eight *P. mume* cultivars from strong-scented, fresh-scented, sweet-scented, and light-scented types were analyzed by headspace solid-phase microextraction/gas chromatography-mass spectrometry. A total of 66 headspace volatiles were putatively detected (Figure 1a), and their relative contents are listed in Appendix A. There were 57, 61, 62, and 56 headspace volatiles corresponding to the strong-scented, fresh-scented, sweet-scented, and light-scented types, respectively. The headspace volatiles included 33 phenylpropanoids/benzenoids, 25 fatty acid derivatives, and 9 terpenoids. The relative contents of various aroma substances in *P. mume* cultivars are shown in Figure 2a. Among them, the relative content of phenylpropanoids/benzenoids was the highest one, which ranged from 88.36 to 96.81%. The relative content of fatty acid derivatives varied from 2.24 to 14.12%, and the content of terpenoids was the lowest, with the relative amount ranging from 0.65 to 3.92%.

A total of 74 endogenous extracts (Figure 1b), including 24 phenylpropanoids/benzenoids, 43 fatty acid derivatives, and 7 terpenoids (Appendix A) tentatively, were identified by organic solvent extraction combined with GC-MS. Among them, 68, 65, 70, and 66 endogenous extracts were detected in strong-scented, fresh-scented, sweet-scented, and light-scented types, respectively. As shown in Figure 2b, the phenylpropanoids/benzenoids were the ingredient with the highest percentage in all the aroma types and varied from 84.77% to 89.73%. Similarly, the relative content of fatty acid derivatives ranged from 7.50% to 10.08%, and the relative content of terpenoids accounted for 2.33% to 5.15%.

### 2.2. The Volatilization Rate of the Main Components of the P. mume Cultivars with Four Aroma Types

Ten components were detected in both headspace volatiles and endogenous extracts in four aroma types. Among them, benzyl acetate, eugenol, benzaldehyde, benzyl alcohol, chavicol, and cinnamyl alcohol were considered to be the main components in most varieties with a relative content of higher than 1% (Appendix A). The volatilization rate was calculated based on the natural logarithm obtained by dividing the content of endogenous extracts by the content of headspace volatiles. The smaller natural logarithm indicates that the volatilization rate is higher. As shown in Figure 3, among the four aroma types, the volatilization rate of benzyl acetate was higher, while that of benzaldehyde was lower. Benzyl acetate has the highest volatilization rate in ‘XLE’ and the lowest volatilization rate in ‘DEZS’. The volatilization rate of benzaldehyde is the highest in ‘DEZS’ and the lowest in ‘MDL’. In addition, the volatilization rate of chavicol in light-scented and sweet-scented types was also higher than that of other scent types. Among them, the volatilization rate of chavicol is the highest in ‘DFH’ and the lowest in ‘DEZS’.

### 2.3. Analysis of Differential Headspace Volatiles of P. mume Cultivars with Four Aroma Types

Based on the relative content of headspace volatiles in eight *P. mume* cultivars, an OPLS-DA model was established, which allowed identifying the differential headspace volatiles of four aroma types as volatile biomarkers. As shown in Figure 4, eight *P. mume* cultivars were divided into four types, and the cross-validated predictive capability (Q2 = 0.776) manifested the model’s feasibility. Moreover, the volatile biomarkers were selected based on the VIP value. As shown in Table 1, 10 differential headspace volatiles with a VIP value greater than 2 were selected as volatile biomarkers to distinguish the four aroma types. The strong and light-scented types were comprised of (*E*)-cinnamaldehyde and camphene. Similarly, the strong and sweet-scented, fresh and sweet-scented types included (*E*)-cinnamaldehyde, (*Z*)-3-hexen-1-ol, and (*Z*)-2-hexenol. Methyl salicylate and camphene were included in fresh and light-scented types. The fresh and strong-scented types were (*E*)-cinnamaldehyde, cinnamyl alcohol. The sweet and light-scented types included benzyl butyrate, pentyl acetate, m-cresol, and (*Z*)-3-hexen-1-ol.

Meanwhile, OAVs were calculated to further decode the relative contribution of volatile biomarkers to the four aroma types. It was found that the OAVs of volatile biomarkers were all greater than 1, including pentyl acetate, (*Z*)-2-hexenol, (*E*)-cinnamaldehyde, methyl salicylate, and their aroma characteristics were mainly fruity, green, spicy, cinnamon (Appendix A). Finally, it can be known that the content of (*Z*)-2-hexen-1-ol in the ‘JM’ of the fresh-scented type was higher than in the other aroma types. Similarly, (*E*)-cinnamaldehyde in strong-scented type was higher than that in other aroma types, and methyl salicylate and pentyl acetate in light-scented type was significantly higher than the ones observed in other aroma types (Appendix A).

### 2.4. Analysis of Differential Endogenous Extracts of P. mume Cultivars with Four Aroma Types

The same method was used to establish the OPLS-DA model of the differential endogenous extracts of *P. mume* cultivars. In this model, a Q2 = 0.694 was determined, showing the reliability of the developed model (Figure 5). As shown in Table 2, 12 differential endogenous extracts with VIP values greater than 2 were used as endogenous biomarkers, which were responsible for distinguishing the four aroma types. Among them, the strong and light-scented types were comprised of tetracosanal and tetracosanoic acid, methyl ester. Similarly, the fresh and strong-scented types comprised 1-heptacosene, hexadecanoic acid, phenylmethyl ester, tricosanal, and chavicol. On the other hand, the sweet and light-scented types included tetracosanoic acid, methyl ester, benzyl benzoate, squalene, and tetracosanal. Tetracosanoic acid, methyl ester, 1-heptacosene, chavicol, and cinnamyl alcohol were included in strong and sweet-scented types. The fresh and sweet-scented, fresh and light-scented types included tetracosanal, cinnamyl acetate, benzoyl cyanide, benzyl benzoate, mandelonitrile benzoate, benzoyl cyanide, tricosanal, and tetracosanoic acid, methyl ester.

At the same time, the greater the OAV of these endogenous biomarkers, the greater their relative contribution to the four aroma types. These endogenous biomarkers with OAV higher than 1 included cinnamyl alcohol and benzoyl cyanide, and their aroma characteristics were floral and fruity (Appendix A). The content of cinnamyl alcohol in the strong-scented type was significantly higher than that in other aroma types, and benzoyl cyanide in the sweet-scented type was higher than those in the other types (Appendix A).

## 3. Discussion

### 3.1. The Similarities and Differences among the Components of Four Aroma Types

The study showed that the main compounds of *P. mume* cultivars from the four aroma types were phenylpropanoids/benzenoids, which is consistent with many ornamental plants, such as *M.*
*ioensis* [13], *Rosa odorata and R. chinensis* [24], *P**. axillaris* [25], and *Luculia pinceana* [26]. Phenylpropanoids/benzenoids accounted for more than 90.57% of the total headspace volatiles and more than 87.71% of the total endogenous extracts. In this study, benzyl acetate, benzaldehyde, benzyl alcohol, and eugenol were used as the main headspace volatiles. In fact, benzyl acetate is also an important aromatic component of many ornamental plants [27,28]. In addition, benzaldehyde was the most abundant endogenous compound among the four aroma types, and the relative content ranges from 27.76 to 46.46%. Additionally, it is a volatile substance that can resist environmental stress in leaves [29]. It is also an important component in other plants, such as *P**. axillaris* [3], *Syringa* [30], and *Bignonia nocturna* [31]. The content of endogenous extracts of the main components in the four aroma types was much higher than that of headspace volatiles; however, the volatilization rates of the four aroma types were different. Therefore, the volatilization of main components of different aroma types may be probably unrelated to the content of endogenous components. Previous studies have shown that volatile substances were transported through the plasma membrane and rely on active transportation. RNA interference down-regulates the *Petunia* pharaonic transporter, which leads to the reduction in volatile matter release [32]. Therefore, the highly volatile components in *P. mume* may be related to the transport mediated by transporters [33]. In addition, the volatilization rates of benzyl acetate were the highest in the four aroma types, which may be due to the higher volatility of ester products, while a variety of alcohols and aldehydes exist at the same time, are difficult to volatilize due to their high polarity. Many aromatic esters are synthesized from the aromatic alcohol and acyl donor catalyzed by the acyltransferase. The rate of aromatic ester biosynthesis is limited not only by the activity of the enzyme responsible for the last step but also by the available substrates. Therefore, understanding the role of the substrate is the direction that needs to be focused on in the future. In addition, compared with previous studies on the floral scent of *P. mume*, this study focused on the differences between headspace volatiles and endogenous extracts among different aroma types and explored the differential components that contributed greatly to the formation of different aroma types, and the number of compounds detected components was higher than that previously described [8,11,18,20,21,22]. Moreover, the in-depth study on the floral scents of different *P. mume* cultivars provides a systematic basis for clarifying the mechanism of the floral scent of *P. mume* cultivars.

In addition, a higher VIP value represents a higher discrimination ability of different types [34,35]. In this study, the volatile and endogenous components with high VIP values of four flavors were studied. The differential components of the four aroma types are different. In this study, 10 differential headspace volatiles and 12 differential endogenous extracts were screened, respectively. Among them, the differential compounds between strong-scented type and other aroma types included (*E*)-cinnamaldehyde and tetracosanoic acid, methyl ester. The differential components of the fresh-scented type contained benzoyl cyanide and tetracosanoic acid. Similarly, the sweet scent-type contained (*Z*)3-hexen-1-ol. Camphene, tetracosanoic acid, methyl ester, tetracosanal, and benzyl benzoate were included in the light-scented type.

### 3.2. Biomarkers Which Have Made Great Contribution to the Four Aroma Types

The floral scent is formed by the interaction of various volatile organic components. In addition, it has been widely accepted that compounds with high OAV contribute more to the aroma of the sample [13,36]. The compounds with higher content and aroma value could be regarded as characteristic components, while the components with lower content and higher odor threshold had a weaker influence on the floral scent [2,37]. In addition, the contents of volatile and endogenous biomarkers and aroma characteristics among the four aroma types were different. Among them, (*E*)-cinnamaldehyde has a cinnamon-like aroma, which makes the aroma more gentle and soft [38]. While cinnamyl alcohol had a floral-like aroma, and its content was higher in the strong-scented type, it is suggested that it may make a significant contribution to the formation of the strong-scented type [39]. The content of (*Z*)-2-hexen-1-ol in the fresh-scented type was significantly higher than that in other types. Previous studies have shown that C_6_, C_9_ aldehydes, and alcohols have fresh and green aromas [40,41], and these aldol compounds were formed by the gradual oxidation, cleavage, and reduction in precursor fatty acids by lipoxygenase (LOX) [42]. Therefore, it is further speculated that they may contribute greatly to the formation of this aroma type. The content of benzoyl cyanide in sweet-scented was higher than in other types. Previous studies have shown that benzoyl cyanide had a bitter almond aroma, which was mainly produced by enzymatic hydrolysis of amygdalin in the fruit [42]. As the most potent contributor, its aroma attributes (fruity odor impressions) may match the sensorial properties of the sweet-scented type. Since methyl salicylate with spicy odor and pentyl acetate with fruity aroma characteristics had higher content in light-scented type [13,43], it is speculated that they may be related to the formation of light-scented type.

In this study, the compounds of four aroma types were identified, and their differential components and contents were analyzed, which provided a theoretical basis for breeding and developing the essential oil of *P. mume*. However, the fragrance of flowers is a comprehensive function of all aroma components. The contribution and aroma characteristics of some compounds are still unknown. Therefore, it is necessary to combine sniffing test with sensory evaluation, including gas chromatography-olfactory (GC-O) and aroma extraction dilution analysis (AEDA) so as to further determine the functions of these compounds and detect the aroma active substances of *P. mume* cultivars and then screen out the key different aroma substances of different aroma types, thus providing a theoretical basis for the breeding of *P. mume* cultivars with different aroma types. Through the continuous study of the floral scent of *P. mume*, the material pool of floral scent in *P.* species and Rosaceae can be expanded, and it is of great significance for studying the evolution of *P.* species and Rosaceae.

## 4. Materials and Methods

### 4.1. Plant Materials

Eight *P. mume* cultivars (Figure 6) with four aroma types were collected in Moshan Mei Flower Garden, Hubei Province (30°33′ N; 114°24′ E) during mid-to-late February 2021, including the strong-scented type: *P. mume* ‘Jiangnang’ (JN), *P. mume* ‘Fenpi Chizhi’ (FPCZ); fresh-scented type: *P. mume* ‘Duoe Zhusha’ (DEZS), *P. mume* ‘Jiangmei’ (JM); Sweet-scented type: *P. mume* ‘Xiao Lve’ (XLE), *P. mume* ‘Longyou’ (LY); light-scented type: *P. mume* ‘Dan Fenghou’ (DFH), *P. mume* ‘Midanlv’ (MDL). Names and aroma types of these cultivars are referred to the book: Chinese Mei Flower (*Prunus mume*) Cultivars in Color (in Chinese press) [19], and the fragrance types also have the preliminary sensory evaluation. Each sample was processed in two ways. One was to cut the branches of flower blossoming and placed them in a box filled with ice, and then instantly transported to the laboratory for headspace analysis of headspace volatiles, and the other was immediately frozen in liquid nitrogen, then stored at −80 °C and then detect endogenous extracts.

### 4.2. Instruments and Reagents

The standard of the *n*-paraffins mixture (C_7_–C_40_) was purchased from ANPEL Laboratory Technologies Inc. (Shanghai, China). Methyl nonanoate was purchased from Sigma-Aldrich (Saint Louis, MO, USA). Methanol of HPLC grade and ethyl acetate were purchased from Fisher Scientific (Fair lawn, NJ, USA), and all other reagents were of analytical grade. DTC-15J Ultrasonic Cleaning Machine was obtained from DingtaiBiochemical Technology Equipment Manufacturing Co., Ltd. (Wuhan, China).

### 4.3. SPME Collection and Solvent Extraction

The volatile compounds were collected by solid-phase microextraction as described in previous studies [18] with minor modifications. In each of three experimental repetitions, 0.2–0.5 g whole blooming flowers were collected and put into 20 mL injection vials, which was safely covered with an aluminum seal and a Teflon septum, with 2.5 µL of methyl nonanoate (0.0875 mg/mL in methanol) was added as the internal standard. The SPME Fiber Assembly 50/30 µm DVB/CAR/PDMS, Stableflex (2 cm) 24 Ga, Manual Holder, 3pk (Gray-Notched) was selected to collect headspace volatiles. The fiber was injected manually and desorbed in the injection port of the gas chromatograph (GC) with helium as the carrier gas. The fiber was desorbed at 250 °C for 5 min in the splitless mode. Before each set of samples was assayed, the fiber was conditioned for 1 h at 250 °C in the injection port of the GC-MS. The empty capped vial was used as the blank control.

As reported in the previous studies [11], the accumulated endogenous components were collected by organic solvent extraction, and some minor modifications were made. In triplicate, approximately 0.40 g powder was placed in a 2 mL centrifuge tube, and 600 μL of ethyl acetate solution containing 0.5 μL/100 mL methyl nonanoate was added as the internal standard. The extraction was performed in an ultrasonication maintained at 4 °C for 40 min and centrifuged at 12,000 rpm for 10 min at 4 °C, and then filtered through a 0.22 µm membrane. The vials were placed on the platform for automatic sample injection.

### 4.4. GC-MS Analysis

The headspace volatiles were analyzed using gas chromatography (Thermo Fisher Scientific, Waltham, MA, USA) onto a fused-silica capillary column (30 m × 0.25 mm i.d., 0.25 µm DB-5MS stationary phase). The endogenous extracts (1 μL) were analyzed by GC-MS equipped with an ISQ (Thermo Scientific, Bellefonte, PA, USA). The injection temperature was kept at 250 °C. The temperature procedure of GC was as follows: the GC oven temperature started at 40 °C, and maintained 2 min, then increased to 250 ℃ by 5 ℃/min, holding for 6 min, and the helium was the carrier gas in the splitless mode. The conditions of the mass spectrometer were as follows: the ion source temperature was set at 230 °C, with electronic impact (EI) mode at 70 eV over the mass range *m/z* 30–300. The solvent cut time was 5 min.

A C_7_–C_40_ alkane standard solution was analyzed regularly to provide references for the calculation of retention time (Kovats) and retention index (RI) and to monitor system performance. Compounds identification was performed by comparing the mass spectra with the NIST 05 (National Institute of Standards and Technology) library and with published data (NIST, http://webbook.nist.gov/chemistry/ (accessed on 3 November 2021)). Semi-quantitative determinations were carried out using methyl nonanoate as an internal standard. The contents of compounds were calculated from the target compounds’ peak areas related to the internal standard peak area.

### 4.5. Odor Activity Values (OAVs)

The OAV of each compound was the ratio of the sample concentration to its respective odor threshold in water. OAV > 1 indicated that the volatile and endogenous biomarkers contribute more to the aroma types [44]. The odor threshold data comes from reported literature data [43,45].

### 4.6. Data Analysis

The results were presented as means of three biological replicates ± standard deviation (SD). SIMCA 14.1, Excel 2016, GraphPad Prism 5, and SAS were used for all statistical analyses. The Heatmap and dendrogram analysis was carried out by ImageGP Version: 1.0 (http://www.ehbio.com/Cloud_Platform/front/ (accessed on 20 July 2020)). The OPLS-DA was applied to discriminate the eight *P. mume* cultivars with four aroma types by using SIMCA 14.1. The VIP value was calculated in the OPLS-DA model, which represented the differences of the variables. When the VIP value was higher than 2, the compounds that were different from each other were screened out as volatile and endogenous biomarkers to distinguish four aroma types. The GraphPad Prism 5 was used to map the relative content of various aroma substances. One-way analysis of variance (ANOVA) with Tukey’s test in SAS software was used to assess volatile and endogenous biomarkers in *P. mume* cultivars with four types.

## 5. Conclusions

In this study, headspace solid-phase microextraction (HS-SPME) and organic solvent extraction (OSE), combined with gas chromatography-mass spectrometry (GC-MS), were used to comprehensively analyze the headspace volatiles and endogenous extracts of eight *P. mume* cultivars with four aroma types. A total of 66 headspace volatiles and 77 endogenous extracts were tentatively identified. Among them, there were six main components in four aroma types, including benzyl acetate (1.55–61.26%), eugenol (0.87–6.03%), benzaldehyde (5.34–46.46%), benzyl alcohol (5.13–57.13%), chavicol (0–5.46%), and cinnamyl alcohol (0–6.49%). Among them, (*Z*)-2-hexenol, (*E*)-cinnamaldehyde, methyl salicylate, cinnamyl alcohol, and benzoyl cyanide may be the reasons for the specific aroma of *P. mume* based on the OPLS-DA and OAVs analyses. This research provides a reference for further enriching the aroma breeding of *P. mume* cultivars.

## Figures and Tables

**Figure 1 molecules-26-07256-f001:**
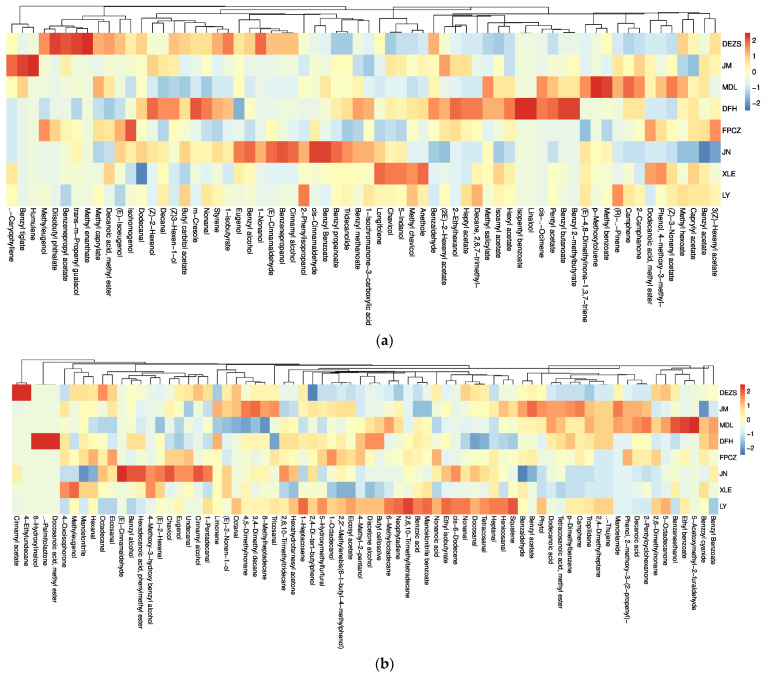
Heatmap and dendrogram analysis of compounds of eight *P. mume* cultivars with four aroma types. (**a**) headspace volatiles; (**b**) endogenous extracts. The color of the heatmap ranges from dark blue (value, −2) to red (value, 2) in the natural logarithmic scale. Data are presented with means of biological replicates. The values were normalized by log_10_ transformation. ‘DEZS’ and ‘JM’ belong to the fresh-scented *P. mume* cultivars; ‘MDL’ and ‘DFH’ belong to the lighted-scented *P. mume* cultivars; ‘FPCZ’ and ‘JN’ belong to the strong-scented *P. mume* cultivars; ‘XLE’ and ‘LY’ belong to the sweet-scented *P. mume* cultivars.

**Figure 2 molecules-26-07256-f002:**
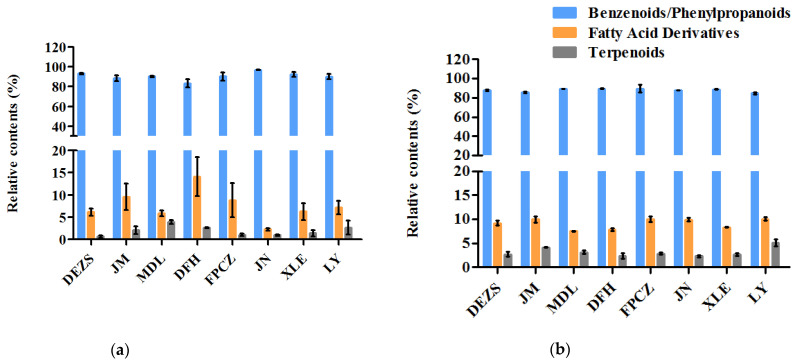
The relative contents of various aroma substances in *P. mume* cultivars with four aroma types: (**a**) headspace volatiles; (**b**) endogenous extracts.

**Figure 3 molecules-26-07256-f003:**
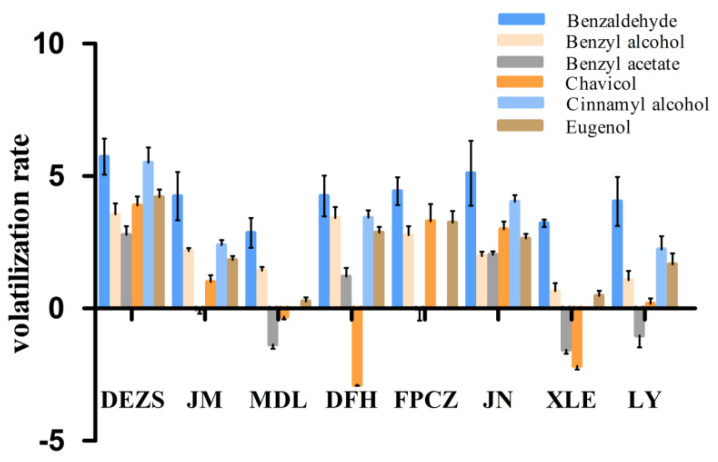
Volatilization rate of main compounds of *P. mume* cultivars with four aroma types. The vertical axis showed the natural logarithms (Ln) of the ratios of the endogenous extracts content to headspace volatiles content.

**Figure 4 molecules-26-07256-f004:**
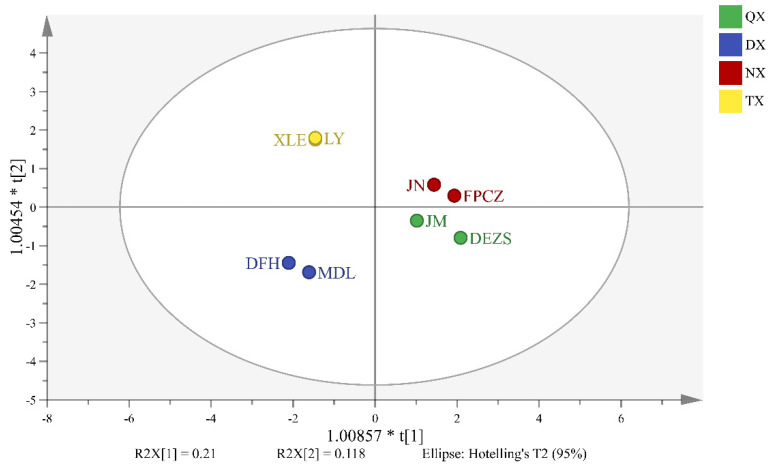
Score plot of OPLS-DA models of headspace volatiles of *P. mume* cultivars with the statistical parameters (R2X = 0.938, R2Y = 0.994, Q2 = 0.776).

**Figure 5 molecules-26-07256-f005:**
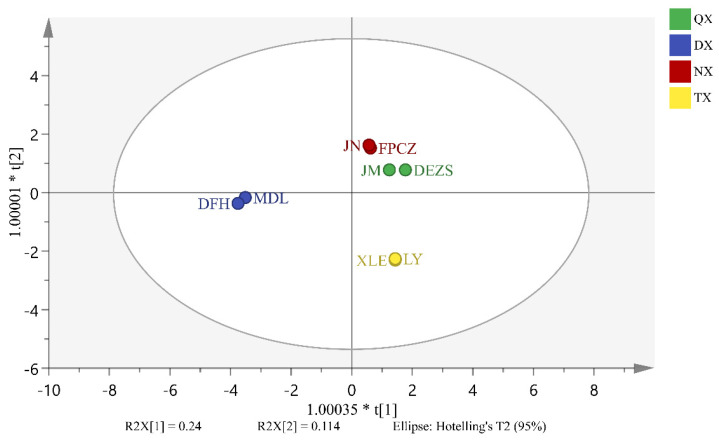
Score plot of OPLS-DA models of endogenous extracts of *P. mume* cultivars with the statistical parameters (R2X = 0.98, R2Y = 0.992, Q2 = 0.694).

**Figure 6 molecules-26-07256-f006:**
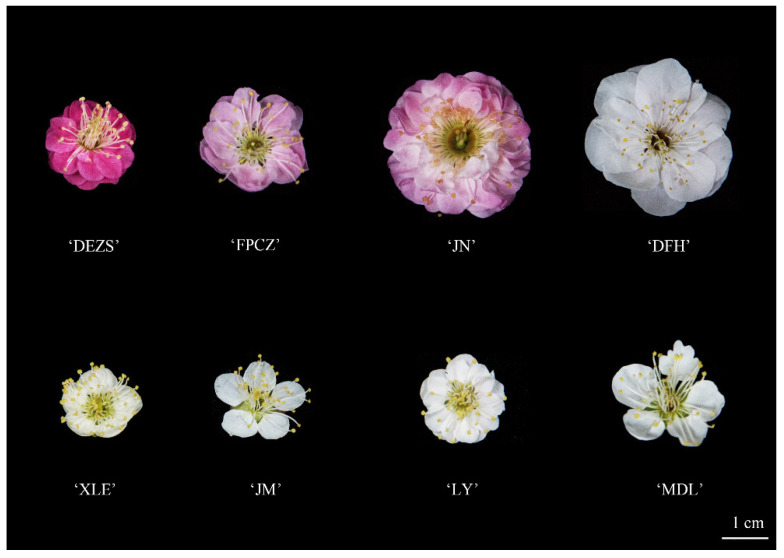
The flowers of eight *P. mume* cultivars with four scent types.

**Table 1 molecules-26-07256-t001:** Volatile biomarkers for the identification of *P. mume* cultivars with four aroma types.

Strong-Scented vs. Light-Scented	Strong-Scented vs. Sweet-Scented	Fresh-Scented vs. Light-Scented	Fresh-Scented vs. Strong-Scented	Fresh-Scented vs. Sweet-Scented	Sweet-Scented vs. Light-Scented
(*E*)-cinnamaldehyde	(*E*)-cinnamaldehyde	methyl salicylate	cis-cinnamaldehyde	(*Z*)3-hexen-1-ol	benzyl butanoate
camphene	(*Z*)3-hexen-1-ol	camphene	cinnamyl alcohol	(*Z*)-2-hexenol	pentyl acetate
			(*E*)-cinnamaldehyde		m-cresole
					(*Z*)3-hexen-1-ol

**Table 2 molecules-26-07256-t002:** Endogenous biomarkers for the identification of *P. mume* cultivars with four aroma types.

Strong-Scented vs. Light-Scented	Strong-Scented vs. Sweet-Scented	Fresh-Scented Vs. Light-Scented	Fresh-Scented vs. Strong-Scented	Fresh-Scented vs. Sweet-Scented	Sweet-Scented vs. Light-Scented
tetracosanoic acid, methyl ester	tetracosanoic acid, methyl ester	tetracosanal	1-heptacosene	mandelonitrile benzoate	tetracosanoic acid, methyl ester
tetracosanal	1-heptacosene	cinnamyl acetate	hexadecanoic acid, phenylmethyl ester	benzoyl cyanide	benzyl benzoate
	chavicol	benzoyl cyanide	tricosanal	tricosanal	squalene
	cinnamyl alcohol	benzyl benzoate	chavicol	tetracosanoic acid, methyl ester	tetracosanal

## Data Availability

The data presented in this study are available on request from the corresponding author.

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
