# Peer review of "Headspace Volatiles and Endogenous Extracts of Prunus mume Cultivars with Different Aroma Types"

_molecules, 2021, doi:10.3390/molecules26237256_

Round 1
Reviewer 1 Report
Thank you for submitting the manuscript “Volatile and Endogenous Compounds of Prunus mume Cultivars with different Aroma Types” to Molecules. Overall, the subject is interesting, but I have some questions.
Lines#43-44: what do endogenous compounds mean? Are volatile compounds not all exogenous as they are released by the material? Perhaps it would be better to name those compounds extracted by headspace as volatile compounds that make up the odor of flowers and the other compounds as volatile compounds since the extraction process is by solvent contact with the sample.
Line#282: please correct Microextraction
Consider adding captions to figures and tables so that they provide information independent of the manuscript text.
Author Response
Responses to Reviewer 1’s comments
We sincerely appreciate your effort in reviewing our manuscript. According to your opinion, we have made corresponding revisions to the Updated Manuscript. Responses to your comments are given below.
Point 1: Lines#43-44: what do endogenous compounds mean? Are volatile compounds not all exogenous as they are released by the material? Perhaps it would be better to name those compounds extracted by headspace as volatile compounds that make up the odor of flowers and the other compounds as volatile compounds since the extraction process is by solvent contact with the sample.
Responses to point 1: Thanks so much for your valuable comments. Actually, the endogenous compounds refer to compounds which are mainly retained in flower tissues and were extracted with ethyl acetate as the extraction solvent. And the volatile compounds are all exogenous as they are released by the plant material. By carefully considering your comments and related literatures [1,2], the endogenous compounds was revised to endogenous extracts while the volatile compounds was revised to headspace volatiles in the whole manuscript, and related description was added in the introduction. (Please see the Introduction (line 41-48))
Reference:[1] Hao R , Du D , Wang T , et al. A comparative analysis of characteristic floral scent compounds in Prunus mume and related species[J]. Bioscience Biotechnology and Biochemistry, 2014, 78(10):1640-1647.
[1]Hao, RJ, Zhang, et al. Emitted and endogenous floral scent compounds of Prunus mume and hybrids[J]. BIOCHEM SYST ECOL, 2014, 2014,54(-):23-30.
Point 2: Line#282: please correct Microextraction
Responses to point 2: In response to your suggestion, we had checked and revised the format of all the ‘Microextraction’ and revised the incorrect ones in the manuscript. (Please see line 349, sub-section 2.1 of Results (line 93))
Point 3: Consider adding captions to figures and tables so that they provide information independent of the manuscript text.
Responses to point 3: Thank you very much for the valuable suggestions. We have added captions to all the figures and tables. (Please see the revised Figures and revised Tables in Supplementary Files)
Reviewer 2 Report
The authors analyzed the of Prunus mume floral aroma, the manuscript is interesting, however, revisions need to be carried out.
in Abstract: add the percentage values ​​of the main compounds identified.
introduction, the authors are talking about aroma that are volatile compounds, The volatiles of plants correspond to a) a
complex mixture of compounds that have the property of naturally volatilizing, under appropriate temperature and humidity conditions, allowing, for example, that we can smell the aroma of a rose, or jasmine. Volatile liquids are essential oils, for example. please be careful with the concepts throughout the introduction it should be reviewed.
Results.
The compound concentration values ​​must appear throughout the text, in addition the chemical composition tables must appear in the MS and not in supplementary materials.
The tables should show the calculated retention rates and those from the literature used in the GC/MS analysis.
Add the compound classes in the tables and their respective concentrations and the total.
Improve the quality of figures.
Discussion
Correct all compound names that are in disagreement, eg (E)-cinnamaldehyde o (E) and (Z) must be in italics.
Materials and methods
Authors must provide the taxonomic identification number of the studied species as well as the name of the taxinomist.
Add the coordinates of the collection location.
What is the reason for not performing quantitative analysis by GC-FID?.
Conclusion
"In this study, a total of 66 volatile compounds were identified by HS-SPME-GC-MS in the eight P. mume cultivars with four types, and 77 endogenous compounds were identified by OSE coupled with GC-MS. Among them, there were 7 main components in four aroma types, but their volatilization rates are different. In addition, 10 and 12 volatile and endogenous biomarkers were also identified when distinguishing the four aroma types through VIP value greater than 2 in OPLS-DA. MS" please relocate for results.
Cite the articles
https://onlinelibrary.wiley.com/doi/10.1002/cbdv.202000982 https://www.mdpi.com/1420-3049/25/4/783
https://www.mdpi.com/1420-3049/25/17/3852
Author Response
Responses to Reviewer 2’s comments
We really appreciate Reviewer 2’s valuable comments for our manuscript entitled “Volatile and Endogenous Compounds of Prunus mume Cultivars with different Aroma Types” (Manuscript ID molecules-1470098). Following your detailed comments, we have made best effort to improve the quality in the Updated Manuscript.
In Abstract:
Point 1: Add the percentage values of the main compounds identified.
Responses to point 1: We have already added the percentage values of the main compounds identified, including benzyl acetate (1.55-61.26%), eugenol (0.87-6.03%), benzaldehyde (5.34-46.46%), benzyl alcohol (5.13-57.13%), chavicol (0-5.46%), and cinnamyl alcohol (0-6.49%). (Please see the Abstract (line 17-19))
Introduction
Point 2: The authors are talking about aroma that are volatile compounds, The volatiles of plants correspond to a complex mixture of compounds that have the property of naturally volatilizing, under appropriate temperature and humidity conditions, allowing, for example, that we can smell the aroma of a rose, or jasmine. Volatile liquids are essential oils, for example. please be careful with the concepts throughout the introduction it should be reviewed.
Responses to point 2: Thank you for your comment. After carefully reviewed the concepts throughout the introduction and referred to some literatures [1,2,3,4,5]. Firstly, the floral scent is a complex of Volatile organic compounds (VOCs). VOCs are composed of a series of lipophilic liquids with low molecular weight, low polarity and high vapor pressure. In addition, the biosynthetic and release pathways of VOCs include a variety of intermediate products. A previous study revealed that endogenous substances were the basis of material volatilization [4]. Before volatilization, the VOCs are produced in liquid form and stored in the tissues of flowers. Due to the change of vapor pressure, a small amount of compounds emitted from flowers and collected by the headspace method are called headspace volatiles. Most of the compounds remained in flower tissues and extracted by solvent are called endogenous extracts[2,4,5]. We have added targeted explanation in the corresponding position of the manuscript according to your comments. (Please see the Introduction (line 33-49))
Reference:
[1]Dudareva, N.; Klempien, A.; Muhlemann, J. K.; Kaplan, I., Biosynthesis, function and metabolic engineering of plant volatile organic compounds. New Phytol 2013, 198 (1), 16-32.
[2] Hao R , Du D , Wang T , et al. A comparative analysis of characteristic floral scent compounds in Prunus mume and related species[J]. Bioscience Biotechnology and Biochemistry, 2014, 78(10):1640-1647.
[3]Naomi, OYAMA-OKUBO, Toshio, et al. Emission Mechanism of Floral Scent in Petunia axillaris[J]. Bioscience, Biotechnology, and Biochemistry, 2014, 69(4):773-777.
[4]Kondo, M.; Oyama-Okubo, N.; Ando, T.; Marchesi, E.; Nakayama, M., Floral scent diversity is differently expressed in emitted and endogenous components in Petunia axillaris lines. Ann Bot 2006, 98 (6), 1253-9.
[5]arman, M.; Mitra, A., Temporal relationship between emitted and endogenous floral scent volatiles in summer- and winter-blooming Jasminum species. Physiol Plant 2019, 166 (4), 946-959.
Results
Point 3: The compound concentration values must appear throughout the text, in addition the chemical composition tables must appear in the MS and not in supplementary materials.
Responses to point 3: Thank you very much for your suggestion. Following your suggestion, we have already added the concentration values of the main compounds identified. (Please see the Abstract (line 17-19), sub-section 3.1 of Disscussion (line 200-201, 204-205), Conclusion (line 360-363)) In addition, the chemical composition tables had also been added in the MS. (Please see the revised Table1 and Table2)
Point 4: The tables should show the calculated retention rates and those from the literature used in the GC/MS analysis.
Responses to point 4: We feel sorry that we didn’t understand the concept of ‘retention rates’. Did you mean the ‘retention index’ or ‘retention time’? Actually, We calculated the retention index (RI) based on the retention time of the C7-C40 alkanes under the same chromatographic conditions. Then, the compounds detected by using GC-MS was identified by comparing their mass spectra with the NIST 05((NIST, http://webbook.nist.gov/chemistry/ ). We have added the RI obtained from the NIST 05 to related tables and the NIST 05 database was also cited. (Please see the revised Table1 to 2)
Point 5: Add the compound classes in the tables and their respective concentrations and the total.
Responses to point 5: Thank you for your excellent comment. We have added the total concentration of the three main compound classes in the chemical composition tables. (Please see the revised Table1 to 2)
Point 6: Improve the quality of figures.
Responses to point 6: The quality of all the figures were improved based on your suggestion. (Please see the revised Figure 1 to 5)
Discussion
Point 7: Correct all compound names that are in disagreement, eg (E)-cinnamaldehyde o (E) and (Z) must be in italics.
Responses to point 7: We sincerely appreciate your careful review. We have corrected the compound names which were inconsistent in the discussion (sub-section 2.3 of Results (line 150,152,154-155,158-159,160,162)), and we have also checked and modified other parts of the manuscript. (Please see the Abstract (line 24), revised Table1-3, sub-section 3.1 and 3.2 of Discussion (line 236,239,248,252,255), Conclusion (line 363) )
Materials and methods
Point 8: Authors must provide the taxonomic identification number of the studied species as well as the name of the taxinomist.
Responses to point 8: Thank you for your careful suggestion. We are very sorry that we have not found the taxonomy identification number and the name of the taxonomist for the naming of the P. mume cultivars currently studied. The naming of the P. mume cultivars were latin name together with the Chinese Pinyin of the cultivars [1,2,3].
Reference
[1] Sun, H.; Zhang, T.; Fan, Q.; Qi, X.; Fei, Z.; Fang, W.; Jiang, J.; Chen, F.; Chen, S., Identification of Floral Scent in Chrysanthemum Cultivars and Wild Relatives by Gas Chromatography-Mass Spectrometry. Molecules 2015, 20 (4).
[2] Zhang, T.; Bao, F.; Yang, Y.; Hu, L.; Ding, A.; Ding, A.; Wang, J.; Cheng, T.; Zhang, Q., A Comparative Analysis of Floral Scent Compounds in Intraspecific Cultivars of Prunus mume with Different Corolla Colours. Molecules 2019, 25 (1).
[3] Rh, A.; Sy, A.; Zz, A.; Yz, A.; Jc, A.; Chen, Q. B., Identification and specific expression patterns in flower organs of ABCG genes related to floral scent from Prunus mume. Scientia Horticulturae 2021, 288.
Point 9: Add the coordinates of the collection location.
Responses to point 9: We had added the the coordinates of Moshan Mei Flower Garden, Hubei Province (30°33′N; 114°24′E) in the Materials and methods part (line 280-281) based on your comment.
Point 10: What is the reason for not performing quantitative analysis by GC-FID?
Responses to point 10: Thank you for your careful review. When quantitatively analyzing the same type of hydrocarbon or organic compounds with large carbon number, GC-FID can obtain more accurate quantitative results without using correction factors. However, the reason why we didn't perform quantitative analysis by GC-FID are as follows. Firstly, GC-MS was widely used for qualitative and quantitative analysis of volatile components in many ornamental plants [7,11,16,17]. Secondly, GC-MS technology is particularly suitable for the analysis of small molecular compound, and the volatile component are lipophilic liquid with low molecular weight and high vapor pressure. In addition, there exists a problem of instrument limitation. Since there is only a mass spectrometer with the detector of gas chromatography in the school experimental platform, and the library of mass spectra (NIST 05) is more convenient to use. Finally, in order to obtain more accurate GC-MS quantitative results, we use the internal standard method to determine the quantitative correction factor. The contents of compounds were calculated from the target compounds’ peak areas related to the internal standard peak area.
Reference:
[1] Baek, Y. S.; Ramya, M.; An, H. R.; Park, P. M.; Lee, S. Y.; Baek, N. I.; Park, P. H., Volatiles Profile of the Floral Organs of a New Hybrid Cymbidium, 'Sunny Bell' Using Headspace Solid-Phase Microextraction Gas Chromatography-Mass Spectrometry Analysis. Plants 2019, 8 (8)
[2] Fan, J.; Zhang, W.; Zhang, D.; Wang, G.; Cao, F., Flowering Stage and Daytime Affect Scent Emission of Malus ioensis "Prairie Rose". Molecules 2019, 24 (13).
[3] Sun, H.; Zhang, T.; Fan, Q.; Qi, X.; Fei, Z.; Fang, W.; Jiang, J.; Chen, F.; Chen, S., Identification of Floral Scent in Chrysanthemum Cultivars and Wild Relatives by Gas Chromatography-Mass Spectrometry. Molecules 2015, 20 (4).
Conclusion:
Point 11: "In this study, a total of 66 volatile compounds were identified by HS-SPME-GC-MS in the eight P. mume cultivars with four types, and 77 endogenous compounds were identified by OSE coupled with GC-MS. Among them, there were 7 main components in four aroma types, but their volatilization rates are different. In addition, 10 and 12 volatile and endogenous biomarkers were also identified when distinguishing the four aroma types through VIP value greater than 2 in OPLS-DA. MS" please relocate for results.
Responses to point 11: Thank you very much for the valuable suggestions. Based on your suggestions, we have rewritten the Conclusion as follows. (Please see the Conclusion (line 356-366)) In this study, headspace solid phase microextraction (HS-SPME) and organic solvent extraction (OSE), combined with gas chromatography-mass spectrometry (GC-MS) were used to comprehensively analyze the headspace volatiles and endogenous extracts of eight P. mume culitivars with four aroma types. A total of 66 headspace volatiles and 74 endogenous extracts were tentatively identified. The main compounds include benzyl acetate (1.55-61.26%), eugenol (0.87-6.03%), benzaldehyde (5.34-46.46%), benzyl alcohol (5.13-57.13%), chavicol (0-5.46%), and cinnamyl alcohol (0-6.49%). Among them, (Z)-2-hexenol, (E)-cinnamaldehyde, methyl salicylate, cinnamyl alcohol, and benzoyl cyanide may be the reasons for specific aroma of P. mume based on the OPLS-DA and OAVs analyses. This research provides a reference for further enriching the aroma breeding of P. mume cultivars.
Point 12: Cite the articles
https://onlinelibrary.wiley.com/doi/10.1002/cbdv.202000982 https://www.mdpi.com/1420-3049/25/4/783
https://www.mdpi.com/1420-3049/25/17/3852
Responses to point 12: The research content of these articles are of great reference significance for us to revise our manuscript, so we added the "https://www.mdpi.com/1420-3049/25/4/783" and "https://www.mdpi. com/1420-3049/25/17/3852" to the section about GC-MS in the introduction (Please see the Introduction(line 60-61)). And "https://onlinelibrary.wiley.com/doi/10.1002/cbdv. 202000982" was added to the section about benzaldehyde in the discussion. (Please see the Discussion (line 208))
Reviewer 3 Report
This manuscript aims to identify the diversity of the floral scent in P. mume cultivars with different aroma types. For this, eight P. mume cultivars were used and analyzed by headspace solid-phase microextraction (HS-SPME) and organic solvent extraction (OSE), both combined with gas chromatography-mass spectrometry (GC-MS). A total of 66 volatile compounds and 74 endogenous compounds were tentatively identified. From these, phenylpropanoids/benzenoids were the main ones. Four aroma types were identified as biomarkers (using VIP values). However, I have several concerns that need to be considered before considering it for publication. One of them is related to the English language and style, and the other is related to the presentation of the results. Detailed comments can be seen in the attached pdf file.

Author Response
Responses to Reviewer 3
This manuscript aims to identify the diversity of the floral scent in P. mume cultivars with different aroma types. For this, eight P. mume cultivars were used and analyzed by headspace solid-phase microextraction (HS-SPME) and organic solvent extraction (OSE), both combined with gas chromatography-mass spectrometry (GC-MS). A total of 66 volatile compounds and 74 endogenous compounds were tentatively identified. From these, phenylpropanoids/benzenoids were the main ones. Four aroma types were identified as biomarkers (using VIP values). However, I have several concerns that need to be considered before considering it for publication. One of them is related to the English language and style, and the other is related to the presentation of the results. Detailed comments can be seen in the attached pdf file.
We really appreciate Reviewer 3’s valuable comments to our manuscript in the attached pdf file. We believe that the changes we have made in the Updated Manuscript according to your comments have made this a significantly stronger manuscript.
Major Concerns:
Point 1: One of the concerns is related to the English language and style of the manuscript.
Responses to point 1: Thanks so much for your valuable suggestion. We have improved our manuscript according to the content of the pdf you modified. Meanwhile, we have also revised the English language and style of other parts of the manuscript. (Please see the “Updated Manuscript”)
Point 2: The other concern is related to the presentation of the results.
Responses to point 2: Thank you very much for your suggestion. Based on the suggestions of you and reviewer 2, we have added Table 1 entitled 'Relative amount of headspace volatiles of eight P. mume cultivars with four aroma types' and Table 2 titled 'Relative content of endogenous extracts of eight P. mume cultivars with four aroma types' (Please see the revised Table1 to 2), and we have modified the presentation of the results.
Others Comments in the attached pdf file:
Point 1: Since the authors have not used pure standards to confirm the identity of the compounds, the term putatively or tentatively should be used.
Responses to point 1: Thank you very much for your suggestion. We have already added the term tentatively to the part of substance identification and other related descriptions in the whole manuscript. (Please see the Abstract (line 16), sub-section 2.1 of Results (line 95,105), Conclusion (line 360))
Point 2: authors have not used chiral columns or pure standards to distinguish among diferent isomers! Notations E and Z should be deleted.
Responses to point 2: We sincerely appreciate your careful review. We had not used chiral columns or pure standards to distinguish among different isomers, however, we tried another method. Since the retention index of substances with the notations E and Z are different. In previous studies, the application of retention index(RI) can better identify isomers or homologues, and improve accuracy [1,2,3,4].
Reference
[1] Yan J , Liu X B , Zhu W W , et al. Retention Indices for Identification of Aroma Compounds by GC: Development and Application of a Retention Index Database[J]. Chromatographia, 2015, 78(1):89-108.
[2]Babushok, V. I . Chromatographic retention indices in identification of chemical compounds[J]. Trends in Analytical Chemistry, 2015, 69:98-104.
[3] Zhu M M. Analysis of volatile compounds and aromatic sunflower seeds and the study of formation mechanism [D]. Anhui Agricultural University, 2014.
[4] Xue S J., Yang J K, Chen S Q., Analysis of Chemical Constitutions of Volatile Oil in Opisthopapus taihangensis by GC-MS Combined with Retention Index. Chinese Journal of Experimental Traditional Medical Formulae: 1-11[2021-11-15]. https://doi.org/10.13422/j.cnki. syfjx.20211864.
Point 3: In my opinion, the 4 aroma types determined should be indicated in the abstract.
Responses to point 3: We have already added a detailed description of the four aroma types in the abstract. (Please see the Abstract (line 26) )
Point 4: In my opinion, a heatmap and dendogram representation of the composition of each sample should be included. This will give a rapid and visual access to the composition of each sample, allowing to see the differences or similarities among them.
Responses to point 4: We really appreciate for your valuable suggestion. Based on the suggestions of you and reviewer 2, we have added Table 1 entitled 'Relative amount of headspace volatiles of eight P. mume cultivars with four aroma types' and Table 2 titled 'Relative content of endogenous extracts of eight P. mume cultivars with four aroma types' (Please see the revised Table1 to 2). And the table contained the relative content of each chemical component. In order to avoid duplication, the heat map and cluster map were not displayed.
Point 5: Please, improve the resolution of Figure.
Responses to point 5: Thank you very much for your suggestion. The quality of all the figures had been improved based on your suggestion. (Please see the revised Figure 1 to 5)
Point 6: The result showed that the smaller the value indicated the greater the volatilization rate. English should be improved!
Responses to point 6: Thank you very much for your suggestion. Following your suggestion, the sentence has been modified to ‘The smaller natural logarithm of indicates that the volatilization rate is higher’. (Please see the sub-section 2.2 of Results (line 130-131)) And the English writing of the whole manuscript had been revised comprehensively.
Point 7: Since methyl salicylate with spicy odor and pentyl acetate with fruity aroma characteristics had higher content in light-scented type [16, 44], it is speculated that they may be related to the formation of this type."This" refers to what type?
Responses to point 7: We are really sorry for making you feel confused. In the original sentence, “this” refers to the light-scented type. And we have added targeted explanation in the corresponding position of the manuscript according to your comment. (Please see the sub-section 3.2 of Discussion (line 267))
Point 8: Please, include the year of collection of the cultivars.
Responses to point 8: Thank you very much for the valuable suggestions. The time of collection of the cultivars was mid-to late February, 2021. And corresponding description had been added in the manuscript. (Please see the sub-section 4.1 of Materials and methods (line 280-281))
Point 9: Standard of n-paraffins mixture (C7-C40) were purchased from ANPEL Laboratory Technologies Inc. (Shanghai, China). The n in n-paraffins should be italic,and 7 and 40 should be inferior to the line.
Responses to point 9: Thank you for your careful review. We have italicized the n in this sentence, and C7-C40 were already inferior to the line (Please see the Materials and methods (line 295)). And all similar issues have been checked and corrected in the whole manuscript. (Please see the Materials and methods (line 332))
Point 10: SPME fiber coated with divinylbenzene/carboxen/polydimenthylsiloxane (50/30 µm DVB/CAR/PDMS) was selected to collect volatile components. The fibre length should be included.
Responses to point 10: The length of the fiber had been added. Based on your comment, we have revised the original sentence as follow “The SPME Fiber Assembly 50/30µm DVB/CAR/PDMS, Stableflex (2 cm) 24Ga, Manual Holder, 3pk (Gray-Notched) was selected to collect headspace volatiles. ” (Please see the sub-section 4.3 of Materials and methods (line 307-308))
Point 11: Do the authors have performed blanks between samples? This should be mentioned.
Responses to point 11: Thank you for your suggestion. We are sorry that we have not described clearly in the Materials and Methods, and a detailed description had been added. Firstly, before testing each group of samples, we treated the fibers at the injection port of gas chromatography-mass spectrometry at 250℃ for 1 h. Secondly, we used the empty vial as a blank control. (Please see the sub-section 4.3 of Materials and methods (line 310-312))
Thank you again for your comments on the revision of our manuscript. We have made corresponding revisions to the pdf content which you modified in Updated Manuscript.
Reviewer 4 Report
Dear Authors,
Despite all advantages, the manuscript contains two, but very important shortcomings: lack of novelty and international actuality.
There are enough of already published papers on the same topic: Refer. 14, 18, 19, 20, 22-24.
The obtained research results could be very actual for Chinese agriculturalists (or in other Asian countries), to the specialists in agriculture and plant breeding domain.
Author Response
Responses to Reviewer 4
We would like to thank the Reviewer 4’s valuable comments for our manuscript. Following your comments, we have made best effort to improve the quality of the Updated Manuscript.
Point: Despite all advantages, the manuscript contains two, but very important shortcomings: lack of novelty and international actuality.
There are enough of already published papers on the same topic: Refer. 14, 18, 19, 20, 22-24.
The obtained research results could be very actual for Chinese agriculturalists (or in other Asian countries), to the specialists in agriculture and plant breeding domain.
Responses to point: Thanks very much for your comments which could improve the quality of our manuscript. By carefully considering your concerns about our research, we have revised our manuscript accordingly. Firstly, we highlighted the innovative points of our research in Introduction (Please see the Introduction (line 67-78)) and subsection 3.1-3.2 of Discussion (Please see the sub-section 4.3 of Materials and methods (line 219-230)). Furthermore, the importance of headspace volatiles and endogenous extracts to elucidate the mechanism of floral scent production were elaborated with more details in subsection 3.1 of Discussion.
However, there are obvious differences between our research and previous studies. Firstly, the focus of our work was to study the headspace volatiles and endogenous extracts released by four aroma types, and further screened out the differential volatile components among the four amora types. Then, the number of compounds detected in this study was higher than that previously described, including 66 headspace volatiles and 74 endogenous extracts. In addition, the research materials we have chosen were different from those of previous studies.
Round 2
Reviewer 2 Report
The authors performed an extensive review of the manuscript, all recommendations were met, I recommend the manuscript for publication.
Author Response
Responses to Reviewer 2’s comments
The authors performed an extensive review of the manuscript, all recommendations were met, I recommend the manuscript for publication.
We are very glad that our manuscript can get your approval, and thank you again for your contribution in the process of reviewing manuscripts.
Reviewer 3 Report
In a global way, the authors have addressed the comments of the reviewers which was translated in a huge improvement of the quality of the manuscript. However, in my opinion, the inclusion of 2 big tables to show the obtained results were not proper. In my opinion, as I have already suggested, a heatmap and dendogram representation of the composition of each sample should be better. These kinds of representations gives rapid and visual access to the composition of each sample, allowing to see the differences or similarities among them, which was not possible to extract when looking for big tables with huge amounts of information. In my opinion, these tables should appear in supplementary materials. Besides, in these Tables, only the RI literature values were present, and no reference to the calculated RIs was made. Without these 2 RI types (RI calc. and RI Lit.) the confirmation of the identifications of the list of detected compounds was not possible.
Author Response
Responses to Reviewer 3’s comments
In a global way, the authors have addressed the comments of the reviewers which was translated in a huge improvement of the quality of the manuscript. However, in my opinion, the inclusion of 2 big tables to show the obtained results were not proper. In my opinion, as I have already suggested, a heatmap and dendogram representation of the composition of each sample should be better. These kinds of representations give rapid and visual access to the composition of each sample, allowing to see the differences or similarities among them, which was not possible to extract when looking for big tables with huge amounts of information. In my opinion, these tables should appear in supplementary materials. Besides, in these Tables, only the RI literature values were present, and no reference to the calculated RIs was made. Without these 2 RI types (RI calc. and RI Lit.) the confirmation of the identifications of the list of detected compounds was not possible.
We really appreciate Reviewer 3’s valuable comments for our manuscript entitled “Volatile and Endogenous Compounds of Prunus mume Cultivars with different Aroma Types” (Manuscript ID molecules-1470098). Following your detailed comments, we have made best effort to improve the quality in the Revised Manuscript.
Point 1: In my opinion, the inclusion of 2 big tables to show the obtained results were not proper. In my opinion, as I have already suggested, a heatmap and dendogram representation of the composition of each sample should be better. These kinds of representations give rapid and visual access to the composition of each sample, allowing to see the differences or similarities among them, which was not possible to extract when looking for big tables with huge amounts of information. In my opinion, these tables should appear in supplementary materials.
Responses to point 1: Thank you very much for the valuable suggestions. We have added a heatmap and dendogram representation of the composition of each sample in the revised Figure1 (a) and (b). (Please see the revised Figure1 (a) and (b)). And according to your suggestions, we had transferred the tables containing the composition of each sample from the text to the supplementary materials. (Please see the revised Tables in Supplementary Files)
Point 2: Besides, in these Tables, only the RI literature values were present, and no reference to the calculated RIs was made. Without these 2 RI types (RI calc. and RI Lit.) the confirmation of the identifications of the list of detected compounds was not possible.
Responses to point 2: Thank you very much for the valuable suggestions. Based on your suggestions, we have already added the calculated RIs in revised Supplementary S1 and S2. (Please see the revised Tables in Supplementary Files)
Reviewer 4 Report
Dear Authors,
I need to mention that some remarkable attempts in order to improve the manuscript, especially emphasizing points of novelty and subtle differences from previously published papers on the same topic have been performed. Despite all these efforts, the manuscript contains the same major shortcomings mentioned earlier.
Author Response
Responses to Reviewer 4
Point: I need to mention that some remarkable attempts in order to improve the manuscript, especially emphasizing points of novelty and subtle differences from previously published papers on the same topic have been performed. Despite all these efforts, the manuscript contains the same major shortcomings mentioned earlier.
Responses to point: Thank you for your recognition of our research. We are sorry that you think our research is not innovative enough, but we still want to explain it. Firstly, P. mume is the only species of Prunus with strong floral scent. Secondly, the material pool of floral scent in P. species and Rosaceae can be expanded, and it is of great significance for studying the evolution of P. species and Rosaceae. Finally, this study mainly focused on the study of P.mume with different aroma types. In the future, we will conduct in-depth research on P.mume with aroma types through a large number of sensory evaluation experiments, and establish an evaluation system for the aroma types. This study was the foundation of future research. And we have highlighted the international actuality of our research in Introduction (Please see the Introduction (line 286-288)) .